# Twist-n-Sync: Software Clock Synchronization with Microseconds Accuracy Using MEMS-Gyroscopes

**DOI:** 10.3390/s21010068

**Published:** 2020-12-24

**Authors:** Marsel Faizullin, Anastasiia Kornilova, Azat Akhmetyanov, Gonzalo Ferrer

**Affiliations:** 1Skolkovo Institute of Science and Technology, 121205 Moscow, Russia; Marsel.Faizullin@skoltech.ru (M.F.); Anastasiia.Kornilova@skoltech.ru (A.K.); A.Akhmetyanov@skoltech.ru (A.A.); 2Software Engineering Department, Saint Petersburg State University, 199034 St. Petersburg, Russia

**Keywords:** clock synchronization, time synchronization, MEMS gyroscope, IMU, angular velocity, smartphone sensors, sensor networks

## Abstract

Sensor networks require a high degree of synchronization in order to produce a stream of data useful for further purposes. Examples of time misalignment manifest as undesired artifacts when doing multi-camera bundle-adjustment or global positioning system (GPS) geo-localization for mapping. Network Time Protocol (NTP) variants of clock synchronization can provide accurate results, though present high variance conditioned by the environment and the channel load. We propose a new precise technique for software clock synchronization over a network of rigidly attached devices using gyroscope data. Gyroscope sensors, or IMU, provide a high-rate measurements that can be processed efficiently. We use optimization tools over the correlation signal of IMU data from a network of gyroscope sensors. Our method provides stable microseconds accuracy, regardless of the number of sensors and the conditions of the network. In this paper, we show the performance of the gyroscope software synchronization in a controlled environment, and we evaluate the performance in a sensor network of smartphones by our open-source Android App, Twist-n-Sync, that is publicly available.

## 1. Introduction

Clock synchronization over a network of sensors is a fundamental component for data gathering and intelligent devices. Emerging fields, such as edge computing, sensor networks, or data fusion, require the highest possible precision in clock synchronization. For example, integration of a global positioning system/inertial navigation system (GPS/INS) needs 1 ms or better synchronization accuracy for less than 5 cm drift in 1 s time horizon [1]. Periodical synchronization with atomic clocks (so-called global clock synchronization) is necessary to overcome this issue.

At the same time, another problem caused by drift arises in the local network of different devices—their timelines should be precisely aligned for sufficient interaction. This task is called *local clock synchronization*. For instance, multi-view camera systems require sub-millisecond synchronization accuracy for capturing high-dynamic scenes [2].

Ideally, a network of sensors can achieve nanosecond and better accuracy if a customized system, dedicated electronics, and triggering signals are introduced. Modern approaches for sensing networks, which include a new plethora of sensors, might not allow for a hardware-based solution, either because of technical issues or geographical reasons, to name a few. Many modern electronics, such as smartphones, provide high-quality data that allows one to utilize them as sensors with wide sensing modalities. However, as products for consumer use, they have no dedicated interfaces for precise hardware clock synchronization. GPS-receivers can also be utilized for clock synchronization by providing Pulse-Per-Second pulses [3], yet this technique is not available in GPS-denied environments. Accordingly, software (or soft) clock synchronization becomes the most desirable solution. Modern smartphones can be viewed as a complex sensors [4], which also require soft clock synchronization.

The widely applied Network Time Protocol (NTP) may provide an accurate solution, but it depends on the communication channel used. Sensor-based approaches, such as image-based synchronization of image streams [5], are also used if specific conditions are maintained.

Our contributions in this paper are the following:We propose a gyroscope-based clock synchronization method that has the following strong points comparing to existing solutions. It provides repeatable microseconds accuracy and precision of synchronization that outperforms other software methods to the best of our knowledge; it needs only a couple of seconds of gyroscopes data for this performance; it is computationally cheap; the method does not need wired connection and can be scaled to any number of systems that need synchronization; and it can be deployed into any general modern smartphone with no additional hardware needed.We develop a software-hardware Multiple Inertial Measurement Unit (IMU) system with precise hardware synchronization for validation of accuracy and precision of our method.We carry out a deep analysis of gyroscope data and configurations that affect the accuracy of the method in order to provide a reference for the optimal choice of parameters.We deploy our method into our open-source publicly available Android Twist-n-Sync App [6], https://github.com/MobileRoboticsSkoltech/twist-n-sync, based on the work of Ansari et al. [7].We create a methodology and setup for estimation of synchronization accuracy and precision of our method on smartphones by using built-in rolling-shutter cameras.

The method has only one limitation: systems must be rigidly connected and twisted for synchronization; after that, they can be detached.

## 2. Related Work

Wired hardware synchronization of sensors by triggering signals or common clock is not often possible or suitable. One can follow alternative approaches of sensors synchronization for this reason. One of these approaches is the utilization of Global Positioning System (GPS) not only as a positioning system but also as a precise global clock [8] that is used for synchronization of sensors. This synchronization is spanning from wireless sensor networks for structural health monitoring [9] to detectors in long-baseline neutrino oscillations experiments [10]. There are methods [11,12] of injection of high-frequency pulses generated by local clocks in between relatively rare GPS Pulse-Per-Second signals in order to get higher clock resolution. These methods are also useful when a loss of GPS-signal appears.

Another important class of synchronization algorithms is network-based protocols; they exchange the time of devices in the network and take into account network transfer time. NTP [13] is the classical protocol implemented on many platforms (desktops/servers/smartphones) designed mostly for global clock synchronization, but it is also actively used in local networks of devices because of its prevalence. The last version can achieve up to 10 ms synchronization accuracy in global networks, and 500 μs and lower in local networks, depending on its’ topology and latency. Other well-developed protocols in that field are Simple Network Time Protocol (SNTP) [14], Precision Time Protocol (PTP) [15], Reference Broadcast Synchronization (RBS) [16], and Lightweight Time Synchronization (LTS) [17], and their modifications demonstrate sub-millisecond performance in ideal conditions but suffer considerably in the case of a lack in stable latency and asymmetric connections.

Since the quality of modern smartphones is continuously improved, these devices are becoming more popular as systems of sensors with a wide range of sensing abilities, and they can provide more benefits when coupled within a network. These types of networks need clock synchronization, as well. Some methods of clock synchronization of a mobile phone network employ smartphone peripherals.

Ansari et al. [7] developed an Android app with their variant of NTP for clock synchronization and frame phases aligning algorithm to get sub-millisecond accuracy in an image capturing on mobile phones. Sandha et al. [18] compared the performance of different synchronization tools available in smartphones (audio peripheral, Bluetooth, WiFi). They state sub-millisecond accuracy using audio and millisecond accuracy using network-based synchronization. Lazik et al. [19] propose a solution for smartphone synchronization to global time by receiving ultrasound from three or more ultrasound beacons with known positions. They achieve 720 μs accuracy of clock synchronization. Benndorf et al. [20] describe a mobile app for smartphones clock synchronization. This app uses audio peripherals to not only listen to ultrasound but also generate it and get ten-millisecond accuracy in their experiments. They mention a limitation of audio-based synchronization expressed in as a small distance between phones as possible. Ahmed et al. [21] solve human coughs detection task by utilizing synchronized acceleration data from a smartwatch with audio from a phone. Their cross-correlation-based synchronization reaches tens of milliseconds error in accuracy.

Shrstha et al. [22] solve the synchronization problem of video records of the same event by analysis of audio tracks. They achieve an accuracy of several milliseconds by audio fingerprinting proposed in Reference [23]. It should be mentioned that all audio-based synchronization methods may suffer from additional time delay that can be explained by the low velocity of sound wave propagation through the air between a speaker and a listener (about 300 μs for 10 cm distance).

Another topic highly related to clock synchronization is image stream synchronization captured from independent cameras, i.e., those cameras that do not have a common triggering scheme. A number of approaches to synchronize images are based on the analysis of common events captured by the cameras. Flashes or any other sharp changes in brightness can perform the role of these events. Shrestha et al. [24] synchronize video recordings by detecting common luminance change of images affected by still camera flashes. This approach provides frame-duration accuracy (about 40 ms). Šmıd et al. [5] utilize similar technique based on flashes detection but in more accurate in-frame scale. The time offset between frames of different cameras estimated by catching a dark-to-bright edge of images where a flash happened. This analysis is possible here due to the rolling-shutter effect of cameras. This method delivers sub-millisecond accuracy. Bradley et al. [25] employ external controlled stroboscope flashes to introduce common exposure for multiple rolling-shutter cameras. Thus, cameras receive reflected light of the scene at the same time. This approach has a limitation: it performs well if there is no background illumination interference with stroboscope lightning. For these approaches above, an external source of light, such as flashes, must be used. In contrast, Caspi et al. [26] propose temporal-spatial matching of images from still cameras by processing trajectories of moving object on a scene and obtain sub-frame synchronization accuracy.

Another issue in clock synchronization is related to clock drift. Because of this phenomenon, one-time synchronization is not enough, and re-synchronization and clock drift compensation is always needed to keep the accuracy of clocks.

Today, atomic clocks with cesium oscillators inside are a standard for precise measuring the passage of time. Despite the existing developments of applying that expensive technology in ordinary life [27], most of the available electronics use RTC (real-time clock) chips with quartz oscillators.This technology is less precise and tends to drift much more over time [28,29]. Clock drift of quartz clocks that are based on quartz crystal oscillators may be originated by a variety of factors, including manufacturing imperfection, temperature, aging, and other effects [28]. The typical values of the drift vary from 1 to 100 PPM (parts per million)—approximately several hundreds of microseconds drift within a minute time duration or several tens of milliseconds in one hour. In a year, the drift becomes a couple of seconds, which can be insufficient for proper application or algorithm work. We briefly address this problem in the paper.

## 3. Background

### 3.1. Rotations and Angular Velocities

Every point belonging to a rigid body has equal angular velocity measured in any common frame of reference. Thus, two rigidly connected ideal gyroscopes will output the same measurements for any instance of time if their axes are aligned. In the case of rotated gyroscopes,
(1)$$\begin{array}{c} \omega_{2}=R_{21}\;\cdot \;\omega_{1}, \end{array}$$
where the values $\omega_{1}$ and $\omega_{2}$ are three-dimensional angular velocities measured by the gyroscopes, and the rotation matrix $R_{21}$ maps the gyroscope measurement from frame 1 to the gyroscope frame 2 [30]. The rotation matrix $R_{21}$ belongs to the group of rotation matrices also known as special orthogonal group SO(3). SO(3) is the set of all 3 × 3 real matrices *R* that satisfy $R^{⊤}R=I$ and detR=1.

If we calculate the L2-norms of the angular velocity $\omega_{2}$, we obtain the following equality (here and below $∥\;\cdot \;∥:={∥\;\cdot \;∥}_{2}$):(2)$$\begin{array}{c} ∥\omega_{2}∥=∥R_{21}\omega_{1}∥={\left(\omega_{1}^{⊤}R_{21}^{⊤}\;\cdot \;R_{21}\omega_{1}\right)}^{\frac{1}{2}}=∥\omega_{1}∥. \end{array}$$

In other words, absolute value of angular velocities for any two rigidly connected frames are equal.

In contrast, accelerations at any point of the rigid body are not equal in their norms because of additional Euler and centrifugal accelerations [31].

### 3.2. Micro-Electro-Mechanical Systems (MEMS) Gyroscopes and Their Imperfections

MEMS gyroscopes are cheap and miniature sensors that are currently massively produced. Due to these advantages, they are broadly used in consumer electronics (smartphones, artificial reality glasses, virtual reality headsets, smart fitness wristbands, image stabilization systems for video cameras), applications in automobiles (automotive roll-over prevention and airbag systems), robotics (pose estimation, sensor fusion, visual-inertial odometry systems [32], abnormally detection), and in other areas.

MEMS gyroscopes still suffer from a number of imperfections [33]. Some of these imperfections are non-unique scaling between measured and true values and misalignment of gyroscope axes. These imperfections are crucial in the influence on the performance of the gyroscopes, and can be well-modeled by
(3)$$\begin{array}{cc} \tilde{\omega } & =SA\;\omega +b+\eta , \end{array}$$
where ω is the true value of angular velocity, $\tilde{\omega }$ is gyroscope measurement, *S* is a diagonal matrix that expresses scaling effect and *A* is an upper unitriangular matrix that corresponds to misalignment effect, b is the slow varying bias, and η is a Winner process. These and other defects of MEMS gyroscopes can be corrected by proper calibration with ground truth data provided via additional tools. There are rotational table-based [34] or visual camera-based [35] methods of calibration among the tools.

### 3.3. Gyroscope Data Conversion

So far, we have considered a gyroscope as a black box that outputs angular velocity measurements over the continuous-time domain. However, most of modern MEMS gyroscopes are digital systems that output discrete data over discrete time. Moreover, to utilize these sensors in an amenable way, their internal structure should be considered. The detailed structure of MEMS gyroscopes and their parameters can be found in Reference [36,37,38]. We will only consider the analog-to-digital conversion of the gyroscope signal because this information is crucial for further analysis.

The signal path of a typical digital MEMS gyroscope from the mechanical part to digital value output can be represented by the scheme in Figure 1. Here, we consider a 1-axis gyroscope for simplicity, the same scheme can be applied to a 3-axis gyroscope with some extensions. The analog voltage from the first block of the scheme, the MEMS angular rate sensor itself, goes through analog anti-aliasing (low-pass) filter to cut high-frequency components of the signal. This is essential to correctly feed it into an analog-to-digital converter (ADC), which is ΣΔ ADC in the majority of modern digital IMUs. The ADC in turn includes three following blocks: ΣΔ-modulator (ΣΔM), digital low-pass filter, and decimator [39]. We will refer to the tandem of two latter blocks as an internal low-pass filter (ILPF) in the next sections.

The ΣΔM generates a high-speed bitstream from the analog signal. The digital low-pass filter along with the decimator are shaping the frequency range of the digital signal and generates a digital signal with a lower output data rate and a higher resolution. This signal is then ready to be transferred by a communication scheme to the microcontroller or microprocessor to be processed. Thus, MEMS gyroscopes output data is characterized not only by the rate and resolution but also frequency range. Because of that, the appropriate treatment of the data must be carried out. For instance, downsampling of given data must be made correctly according to Nyquist-Shannon sampling theorem [40].

### 3.4. Gyroscope Measurements

Although angular velocities are time-continuous values, digital gyroscopes convert measurements of angular velocities into a three-dimensional function of discrete time. We denote this function by a column vector $\omega_{i}\left[t_{j}^{n}\right]$ as a measurement of the gyroscope *i* at the instance of time $t^{n}$ with integer index *n* of time clock *j*. Square brackets highlight the discrete nature of the argument. For simplicity, we will index the time clock index related to a specific gyroscope by using the same index. Thus, for *i*-th gyroscope, we have $\omega_{i}\left[t_{i}^{n}\right]$.

## 4. Clock Synchronization Using Gyroscopes

Let us first formalize the problem of clock synchronization we aim to solve in this work:

**Problem** **1.**
*Consider two independent clock systems $t_{1}$ and $t_{2}$, each of which measures the absolute time τ. The problem is to find the relative time delay $\Delta t_{21}\left(\tau \right)$ such that:*
(4)$$t_{1}\left(\tau \right)+\Delta t_{21}\left(\tau \right)=t_{2}\left(\tau \right).$$


This formal definition is impractical for the following reasons: (i) Absolute time τ is not observable, and (ii) these are stochastic processes conditioned by a multitude of factors. Because of this, we will follow a more pragmatic solution, which consists of estimating this time delay at a particular instant of time, i.e., observing a common event from both systems. Then, one can estimate $\Delta t_{21}\left(\tau_{event}\right)$ by observations of the same event $z(\tau_{event})$. From now on, we will omit the current absolute time $\left(\tau =\tau_{event}\right)$ and refer to the relative time delay of both observations as
(5)$$z_{1}\left(t_{1}+\Delta t_{21}\right)=z_{2}\left(t_{2}\right).$$

This technique of finding common events has been used since ancient times: the rise and fall of the sun to set a new day or the zenith of a constellation marking the pass of the seasons. In modern days, the required accuracy has dramatically increased.

We propose clock synchronization based on measurements of angular velocities $\omega_{i}\left[t_{i}^{n}\right]$ by two gyroscopes that belong to the same rigidly connected body (the number of gyroscopes can potentially be increased). This process involves an analysis of data in both discrete and continuous-time domains to, firstly, obtain a coarse estimation $\Delta t_{21}^{init}$ of delay $\Delta t_{21}$ and, secondly, perform a refinement $\Delta t_{21}^{final}$ of this estimation. The diagram of our approach is shown in Figure 2 and described below in this section.

We analyze gyroscopes data in sets of trials that correspond to movements in three stages: *stay still–twist–stay still*. The trials last about 5 s in average. Below, we describe our method.

### 4.1. Coarse Time Delay Calculation

From now on, we analyze a single trial of measurements for every gyroscope. This step is required for obtaining an initial estimation of the delay $\Delta t_{21}^{init}$ between timestamps of measurements of a pair of gyroscopes. This is done by calculation and analysis of discrete-time cross-correlation *C* of absolute values of measured angular velocities
(6)$$\begin{array}{c} C\left[k\right]=\sum_{n=1}^{N}∥\omega_{1}\left[t_{1}^{n}+kT\right]∥\;\cdot \;∥\omega_{2}\left[t_{2}^{n}\right]∥, \end{array}$$
where *T* is the sampling period of measurements, *k* is an integer index, *N* is the number of samples, and $\omega_{1}\left[t_{1}^{n}+kT\right]=0$ for n+k∉[0,N].

The initial delay estimation is related to the index of the maximum value of cross-correlation:(7)$$\begin{array}{c} \hat{k}=\arg\max_{k}C\left[k\right] \end{array}$$
by the equation
(8)$$\begin{array}{c} \Delta t_{21}^{init}=T\;\cdot \;\hat{k}. \end{array}$$

### 4.2. Online Calibration

We avoid the full gyroscope calibration (Equation 3) since it is not strictly necessary for our method. Instead, we utilize the relative calibration of gyroscopes. Thanks to this approach, there is no need for external calibration tools and ground truth values, neither for angles nor angular velocities.

We also consider time-dependent biases, introduced in the gyroscope model (Equation 3), to be constant because the characteristic time of change of their values is usually larger than the duration of the synchronization sequence (several minutes versus several seconds [41]). We subtract these constant biases beforehand and exclude them in further steps. Gyroscope biases can also be affected by temperature [33]. However, we assume that the temperature stays constant during every single trial.

After the initial estimation of time delay, we introduce relative calibration of resulted measurements with time stamping corrected by (Equation 8).

Using (Equation 3) and (Equation 1), we can relate gyroscopes data by
(9)$$\begin{array}{cc} {\tilde{\omega }}_{2} & =S_{2}A_{2}R_{21}A_{1}^{-1}S_{1}^{-1}\left({\tilde{\omega }}_{1}-\eta_{1}\right)+\eta_{2}=M{\tilde{\omega }}_{1}+\xi , \end{array}$$
where the matrix $M=S_{2}A_{2}R_{21}A_{1}^{-1}S_{1}^{-1}$ is the relative correction between gyroscopes, and $\xi =\eta_{2}-S_{2}A_{2}R_{21}A_{1}^{-1}S_{1}^{-1}\eta_{1}$ is the new noise vector. We assume that, for analogous gyroscopes, $\eta_{1}∼N\left(0,\alpha I\right)$, $\eta_{2}∼N\left(0,\alpha I\right)$ with some positive scalar number α and identity matrix $I\in R^{3\times 3}$. Thus, we model ξ∼N(0,βI) with scalar β because *M* is close to orthogonal matrix.

Then, the optimal solution for linear mapping in (Equation 9) in terms of minimum mean square error, can be found by
(10)$$\begin{array}{c} M^{opt}=\arg\min_{M}\sum_{n=1}^{N}{∥\omega_{2}\left[t_{2}^{n}\right]-M\;\cdot \;\omega_{1}\left[t_{1}^{n}+\Delta t_{21}^{init}\right]∥}^{2}. \end{array}$$

The solution to the least squares problem (Equation 10) leads to the following closed-form:(11)$$\begin{array}{c} M^{opt}=\Omega_{2}^{T}\Omega_{1}{\left(\Omega_{1}^{T}\Omega_{1}\right)}^{-1}, \end{array}$$
where $\Omega_{i}={\left[\begin{array}{ccc} \omega_{i}\left[t_{1}\right] & \cdots & \omega_{i}\left[t_{N}\right] \end{array}\right]}^{T}$ is N×3 matrix, stacked from all the measurements.

Now, steps (Equation 6), (Equation 7), (Equation 8) are repeated to obtain new cross-correlation and determine new initial guess of delay with new values of $\omega_{1}:=M\omega_{1}$.

### 4.3. Time Delay Refinement

The coarse optimization of the time delay proposed in Section 4.1 is an exhaustive search over the index *k* that provides the interval of C[k], where the global optimum is. One can improve this initial estimate by properly using optimization techniques over the Time-Continuous (TC) correlation function C(p), defined below.

To this end, we analyze the interpolated data C(p) of the cross-correlation signal C[k]. Interpolation is done by natural cubic spline [42]. Thus, the interpolation of the discrete-time cross-correlation is computed as
(12)$$\begin{array}{c} C\left(p\right)=\left\{\begin{array}{cc} C_{1}\left(p\right), & \mathrm{if}\;1\leq p<2\\ \ldots & \\ C_{k}\left(p\right), & \mathrm{if}\;k\leq p<k+1\\ \ldots & \\ C_{N-1}\left(p\right), & \mathrm{if}\;N-1\leq p\leq N \end{array}\right.. \end{array}$$

Each part of the function C(p) corresponds to the third order polynomial:(13)$$\begin{array}{c} C_{k}\left(p\right)=C\left[k\right]+a_{k}\left(p-k\right)+b_{k}{\left(p-k\right)}^{2}+c_{k}{\left(p-k\right)}^{3}, \end{array}$$
where $a_{k}$, $b_{k}$, $c_{k}$, along with C[k], are coefficients of the spline in this interval. The domain of the TC correlation is defined over the interval p∈[1,N] of real values.

In order to calculate the global optimum, we do the following assumption, which always holds according to our experiments:

**Assumption** **1.***The global maximum of the interpolated cross-correlation C(p), defined by (Equation 12) and (Equation 13), lies within the interval $\left(\hat{k}-1,\hat{k}+1\right)$ around the* coarse solution *from (Equation 8)*.

This assumption is natural because (i) Equation (Equation 6) is a sufficiently accurate estimate that differs from the true delay by less than a sampling period *T* and (ii) a cubic spline does not suffer from Runge’s phenomenon [43], and this alleviates the effect of undesired oscillations present in other interpolation techniques.

Due to this assumption, the search of the refined global maximum time delay $\Delta t_{21}$ is needed only within the union of intervals where $C_{\hat{k}-1}\left(p\right)$ and $C_{\hat{k}}\left(p\right)$ are defined. By construction, C(p) is a continuously differentiable (smooth) function, and the derivative of the *k*th spline part is
(14)$$\begin{array}{c} \frac{dC_{k}\left(p\right)}{dp}=a_{k}+2b_{k}\left(p-k\right)+3c_{k}{\left(p-k\right)}^{2}. \end{array}$$

For this optimization, we solve the quadratic equations of the first-order derivatives of these two splines to determine all the local extrema within the intervals $\left(\hat{k}-1,\hat{k}\right)$ and $\left[\hat{k},\hat{k}+1\right)$. The first-order necessary condition for optimality $\frac{dC_{\hat{k}}\left(p\right)}{dp}=0$ for the second interval gives
(15)$$\begin{array}{c} a_{\hat{k}}+2b_{\hat{k}}\left(p-\hat{k}\right)+3c_{\hat{k}}{\left(p-\hat{k}\right)}^{2}=0. \end{array}$$

This quadratic equation has two roots $p_{1_{\hat{k}}}$ and $p_{2_{\hat{k}}}$. Any root that does not belong to the second interval is not considered as a candidate to be the potential solution. The same procedure is done for the first interval. After that, we obtain up to four candidates to be the final estimation of the refinement. We choose only one of them that gives the global maximum of spline (Equation 13). We denote this solution by $\hat{p}$.

Figure 3 shows the position of $\hat{p}$ among the locality of $\hat{k}$. Finally, the estimated delay between two clock systems is
(16)$$\begin{array}{c} \Delta t_{21}^{final}=T\;\cdot \;\hat{p}. \end{array}$$

The method is computationally cheap because the most complex operation, computation of discrete-time cross-correlation, employs Fast Fourier Transform implementation (O(NlogN) [44]). The other steps of the algorithm are meaningless in terms of computational load.

The approach can be extended to synchronize three or more systems equipped with gyroscopes by simply assigning pairs of measurements, for instance, from one primary gyroscope to all others.

## 5. Experiments on Controlled Conditions: Data Acquisition Platform

In this first stage of the experiments, we have recreated conditions where it is possible to measure a ground-truth delay in order to correctly compare our gyroscope-based synchronization.

To carry out the experiments and evaluate our approach, we have developed a handheld inertial sensor system consisting of two IMUs MPU9150 and a developers board based on the microcontroller unit (MCU) STM32F407. Accelerometer data are not used. These IMUs are widely used in modern electronics and do not have any specific features. Any other IMUs can be used instead of them. We only utilize external clock for convenience that can also be excluded. We set up the lower full scale range of angular velocity to get the highest resolution of measurements. All sensors are rigidly attached to the metal platform, as shown in Figure 4a. The MCU performs data collection from both IMUs, and all the IMU sensors are synchronized on the hardware level and have a shared quartz clock generator. The ground-truth delay is measured by an oscilloscope as the time between “Data ready” signals from both IMUs with 40 ns resolution.

Gathered data, along with measured initial delay, is sent to PC with a sample rate of 1000 samples per second to be saved and utilized for examination of MEMS gyroscope-based clock synchronization accuracy and precision (CSAP). The block-scheme of the system is depicted in Figure 4b.

To verify our approach, we collected two series of 277 and 209 trials of movements of our handheld system—the first series with a high and the second with a low cut-off frequency of the IMUs. Details of the trials are stated in Section 4.

To widely examine CSAP, we divided data treatment into several domains. The domains are described below and include influence on CSAP by calibration, the choice of ILPF cut-off frequency, downsampling, and the shift between downsampled measurements.

We utilize Median Absolute Error (MedAE) and Interquartile Range (IQR) of absolute error as a numerical measure of CSAP in this section.

### 5.1. Calibration Influence

The calibration of sensors, as explained in Section 4.2, is the crucial stage of such a type of clock synchronization. We have found that this process improves MedAE of CSAP 9.5 times and IQR more than 11 times comparing to raw measurements. This comparison can be seen in Table 1. There are also Mean Absolute Error (MAE) and Standard Deviation (SD) depicted. According to the table, even uncalibrated measurements provide sub-millisecond CSAP.

### 5.2. Downsampling and Shift Influence

This section is related to the trade-off between CSAP and the amount of data of IMUs to be processed. For that, we have created a set of downsampling factors ranging from 1 kHz down to 5 Hz.

In order to decrease the amount of data and, consequently, the computational power, one may utilize downsampled measurements instead of data with the original high sample rate. However, this may lead to the loss of useful information about dynamics in data. Additionally, the loss of information can lead to the degradation of the CSAP. In our experiments, we have applied downsampling of data from 1 kHz down to 5 Hz drawn by several values of downsampling factors to better quantify their degradation and to select the ideal downsampling factor before losses are noticeable or severe.

We examine two extreme cases of the time shift between different gyroscope instants: perfect alignment of data frames and the maximum T/2 time distance between them. These two cases are depicted in Figure 5. Perfect alignment has the best CSAP because the measurements made at the same instances of time differ only because of the noise and imperfections of other sensors. In contrast, the half-of-period shift has the worst CSAP because of the distinction of measurements also affected by the dynamics of movements that may change a lot during this horizon. We also assume that CSAP of any other case of alignment lies in between these extreme cases.

Error dependence on the downsampling factor is shown in Figure 6. Error for half-of-period shift begins to rise fast starting from downsampling factor of 4 (250 Hz) and reaches three orders in accuracy and four orders in precision for the highest downsampling factor. Error for zero shift case also increases but slower and goes up to about five times in accuracy and 100 times in precision.

High values of downsampled data error can also be explained by incorrect data treatment, namely the absence of essential anti-aliasing filtration step before downsampling. Thus, data should be filtrated before downsampling, or a small value of downsampling factor should be chosen if no filtration is applied. For our case, downsampling down to 250 Hz with no additional filtration keeps the CSAP within the same order.

The influence of different cut-off frequencies of ILPF on CSAP is considered in the next subsection.

### 5.3. Cut-Off Frequency Influence

ILPF of some gyroscopes may be parametrized to have different shapes of frequency response. One of the key parameters of this parametrization is cut-off frequency. Higher cut-off frequency provides more information about the dynamic of movements but introduces more noise into measurements.

The IMUs of our system allow parameterization of internal low-pass filter (ILPF) by a grid of cut-off frequencies. We utilize this feature and pick up two extreme available setups, the lowest (5 Hz) and the highest (188 Hz) frequencies in the experiments.

Dependence of error on two values of cut-off frequency for several downsampling factor values is depicted in Figure 7. This figure shows that the growth of the error for different cut-off frequencies has different patterns. In particular, better accuracy of delay estimation on low downsampling factors is achieved with the measurements gathered with high cut-off frequency. However, measurements with narrower bandwidth provide better performance starting from downsampling factor 8 (125 Hz), and this superiority stays the same till the highest factors.

This behavior on the high downsampling factors can be explained by overlapping the data frequency range on half of the downsampled sample rate for the case of the high cut-off frequency. In contrast, low cut-off frequency resolves the overlapping problem but shows the lack of available useful information on high sample rates comparing with high cut-off frequency.

### 5.4. Overall Recommendation

The overall recommendation on the application of IMUs is to choose ILPF-parameters (mainly cut-off frequency), which should be necessary and sufficient for catching high-frequency components of movements and select the data rate accordingly. If there are no available parameters, then software anti-aliasing filtration, followed by appropriate downsampling, should be applied. The recommendation holds not only for clock synchronization but also in motion estimation algorithms used in inertial navigation and any other analogous problems.

### 5.5. On the Duration of Trials

In Figure 8, statistics on error dependence on the trials duration are shown. The minimum and the maximum duration here are 2.45 and 10.9 s, accordingly. The error values are extended from 0.12 to 168.20 μs. From this point cloud, it can be stated that the trial duration within this range does not affect error values much.

## 6. Experiments on Uncontrolled Conditions: Smartphones

The setting with uncontrolled conditions includes all those devices that cannot control the time delays by a precise signal, mostly smartphone devices; therefore, it is impossible to measure the synchronization quality directly. Still, this section will discuss the methodology we have followed to demonstrate how gyro-based synchronization can be applied to image capture synchronization and what synchronization quality can be achieved. Firstly, we overview an existing Android app for synchronous image captures from several smartphones based on the NTP protocol, and then we describe its adaptation for our gyro-based synchronization. After that, we propose a method for evaluating synchronization quality using our developed LED flashes and rolling-shutter effect and compare the performance of NTP-based synchronization with our gyro-based synchronization, showing indisputable advantages of the latter.

All experiments were conducted on two Samsung S10e smartphones, CPU: Samsung Exynos 9820 2.73 GHz 5.4 Gb RAM with built-in MEMS IMU sensor LSM6DSO. IMU sensor supports up to 6 kHz output sample rate, but only 500 Hz is available in Android; therefore, we use this frequency in our experiments. Samsung S10e provides two back cameras–standard and wide-angle. We use the second one for our task as it has less pre-processing features that can affect measurement quality (i.e., multi-frame noise reduction).

### 6.1. Android Application for Synchronous Image Capture

To demonstrate the quality of gyro-based synchronization on smartphones, we use and modify open-source Android application libsoftwaresync released by Google Research [7], which allows users to synchronously capture photos on multiple smartphones.

The proposed approach has two main steps (Figure 9): (i) perform clock synchronization between smartphones using a modified NTP version and (ii) start continuous image streaming on devices and align capture phases.

The application requires a user to start a Wi-Fi hotspot network on one device, acting as a leader smartphone and controlling other devices running the application in its network. Then, the leader and the clients perform an NTP handshake by exchanging synchronization messages. The leader smartphone runs the clock filter algorithm to select from multiple NTP samples and returns computed offset from its local clock to the client. In general, the main disadvantage of network-based protocols and, particularly, NTP is that they strongly depend on the network latency and symmetry; that is why their best accuracy can be achieved only in a local low-load network in isolated conditions.

The next step is to perform synchronous capture, which poses a separate problem. This is because the latency between a frame software request and the smartphone sensor exposure is highly variable. To solve this problem, libsoftwaresync requests continuous capture session with constant frame rate and aligns NTP synchronized capture phases of all devices to a predefined goal phase (Figure 10).

### 6.2. App Modification with Gyro-Based Clock Synchronization Algorithm

To evaluate the developed algorithm, we modified the clock synchronization module of libsoftwaresync—instead of the NTP synchronization step, we use gyroscope-based synchronization (Figure 11). Firstly, by pressing the “Synchronization” button, our modification starts recording gyroscope values with timestamps using predefined frequency on leader and client devices. During the synchronization period, the user should rigidly attach smartphones that could be effortlessly done by holding them in one hand. When this step is completed, recorded data from all smartphones are gathered on the leader device and sent to the Python server with a gyroscope clock synchronization algorithm for processing. The server executes gyroscope data calibration and calculates the offset via our method (Section 4). The offset is returned first to the leader and then to the client’s smartphone. After the client receives the offset, the rest of the capture process is identical to that of libsoftwaresync.

### 6.3. LED Setup for Evaluation

To estimate the time delay between two image frames of smartphones, we use a setup with a light source (led strip lights) covering almost the whole camera fields of view. It blinks with a period of 16 ms (8 ms light is off, 8 ms light is on). We capture this light source and utilize the rolling shutter property of CMOS camera matrices, where rows (lines) are capturing sequentially from top to down with time delay between the neighboring lines defined by one line readout time [5]. Due to this fact, one can observe bright and dark segments on the obtained images (Figure 12). For Samsung S10e, one line readout time is approximately 9 μs. Thus, we define the delay between frames from different smartphones as the difference between dark-to-bright edge line numbers multiplied by one line readout time (Figure 13).

We employ the readout time as the resolution of our measurements. Exposure time is 1/16,000 s and does not affect the resolution because any exposure of single line is enough for visible distinguishing of dark-to-bright edge of obtained images.

Due to our setup, the positions of cameras do not have to be aligned at all because the delay is encoded in the line numbers. In contrast, Reference [7] requires an external tool for measurements that must be examined and post-processed for better resolution, which is a more complicated process than our technique.

### 6.4. Results

To evaluate the clock synchronization algorithms, we considered the following setups with two smartphones: (i) Google Research app with NTP synchronization in the local network, (ii) Google Research app with NTP synchronization and general data transferring in the background by other apps, and (iii) modified app with gyro-based synchronization with data transferring in the background by other apps. Comparison results are presented in Table 2. The NTP-based algorithm results in the local network are close to the ones stated in the original paper. In the network load case, we see that the NTP algorithm suffers considerably and becomes less stable, whereas the gyro-based algorithm demonstrates stable accuracy, even better than NTP, without network load. The result of our gyroscope-based algorithm also stays in accordance with the results obtained for the controlled system (Section 5).

### 6.5. Demonstration

This part of our experiments contains a demonstration of synchronized images taken by two smartphones after applying our gyroscope-based clock synchronization method. These photos capture highly-dynamic actions in order to challenge the synchronization performance of our method. The smartphones are held in the hand and have about 5 cm of displacement (mostly horizontal) of the built-in cameras. Every camera has 1/1000 s exposure time. This qualitative experiments are depicted in Figure 14. The pair of images with 33 ms offset is also shown there for comparison. For more information, please check the project site [6].

### 6.6. Necessity of Periodical Re-Synchronization

In this part, we briefly examine the relative clock drift of the smartphones used in the experiments. A synchronized sensor network will eventually lose synchronization due to the clock drift, as mentioned in Section 1.

In order to measure the relative clock drift of our smartphones, we carried out experiments of periodic clock synchronization by our method for 11 min. We recorded 127 trials of twisting the smartphones. Every twist lasts about 5 s.

The measured offset between two clocks is depicted in Figure 15. Estimated clock drift reached 6.5 ms during the experiment or about 9.5 ppm, in accordance with typical values of drift.

For such values of drift, periodical re-synchronization of the smartphones is needed, with about a 2-min period, in order to keep sub-millisecond accuracy of synchronization. If sensors are continuously moving naturally (structure from motion, visual odometry), these extra-movements could be reused for clock synchronization, as well. Re-synchronization can be done online, but this is beyond the scope of the paper.

The re-synchronization period may be significantly increased if the time drift can be compensated by other techniques (e.g., Reference [45]).

## 7. Further Work

We employ relative calibration in our method; however, better performance may be achieved due to the complete calibration of inertial sensors. Because of the time drift of independent clocks, our algorithm should be implemented along with other existing techniques for clock drift compensation. The current smartphone app involves an external server for offset computations, but these actions can be moved to the leader smartphone in future work. For simplicity, our modification currently supports configuration with only two smartphones that can also be extended to more devices without synchronization performance degradation.

## 8. Our Contribution

We have developed a method for software clock synchronization based on the measurements of MEMS-gyroscopes, and we have examined its performance on our designed multiple-IMU hardware system. In addition, an in-depth analysis of gyroscope sensor parameters has been carried out, and, according to our report, it is best to select high frequencies with relatively small decimation factors (250Hz and higher) while setting up the IMU cut-off frequency in accordance with Nyquist-Shannon sampling theorem.

We have developed a setup using a source of light and the methodology for measuring the accuracy and precision of our algorithm on smartphones. We have integrated our algorithm into an existing Android app in order to demonstrate the synchronization performance. Our method outperforms the original SNTP-based algorithm, achieving an accuracy of several microseconds.

## Figures and Tables

**Figure 1 sensors-21-00068-f001:**
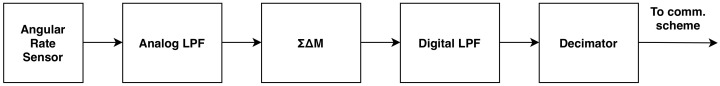
Block scheme of a 1-axis MEMS digital gyroscope.

**Figure 2 sensors-21-00068-f002:**
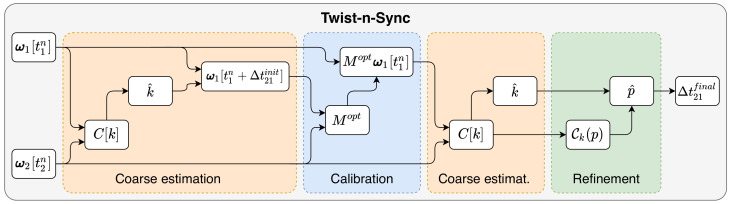
Twis-n-Sync algorithm diagram. The algorithm has raw angular velocity measurements from two rigidly attached gyroscopes as the input data and outputs an estimated offset between the timestamps of the measurements. The algorithm contains four stages: the first coarse estimation, calibration, the second coarse estimation, and refinement. If the sensors are previously calibrated, than two the first stages can be skipped.

**Figure 3 sensors-21-00068-f003:**
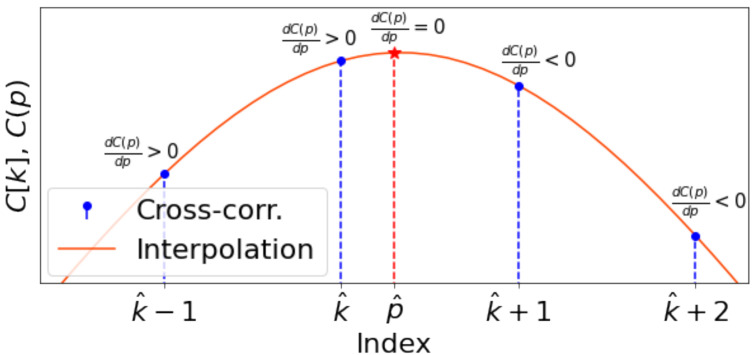
Cross-correlation and its interpolation by cubic spline in local area of $\hat{k}$. Refinement $\hat{p}$ lies within the interval $\left(\hat{k},\hat{k}+1\right)$ in this case.

**Figure 4 sensors-21-00068-f004:**
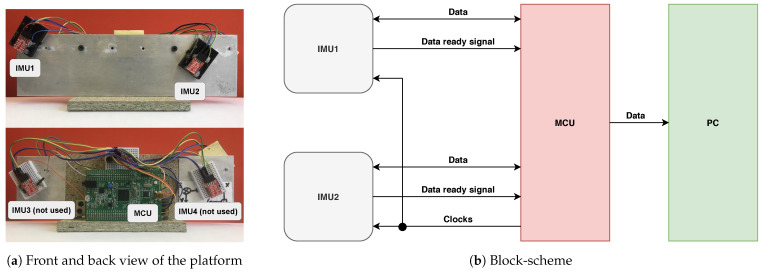
(**a**) Common view of gyroscope-based clock synchronization accuracy estimation system. Only two IMUs and MCU board are used. (**b**) Block scheme of the system. Signals transfer direction denoted by arrows directions.

**Figure 5 sensors-21-00068-f005:**
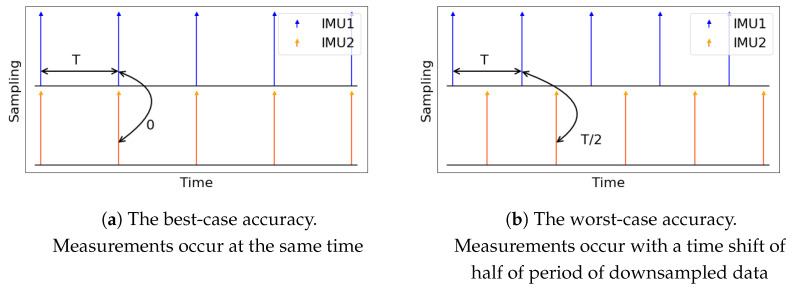
Two extreme cases of the shift between the nearest downsampled measurements of two IMUs. Accuracy of every other case is assumed to be between the best-case and worst-case accuracy

**Figure 6 sensors-21-00068-f006:**
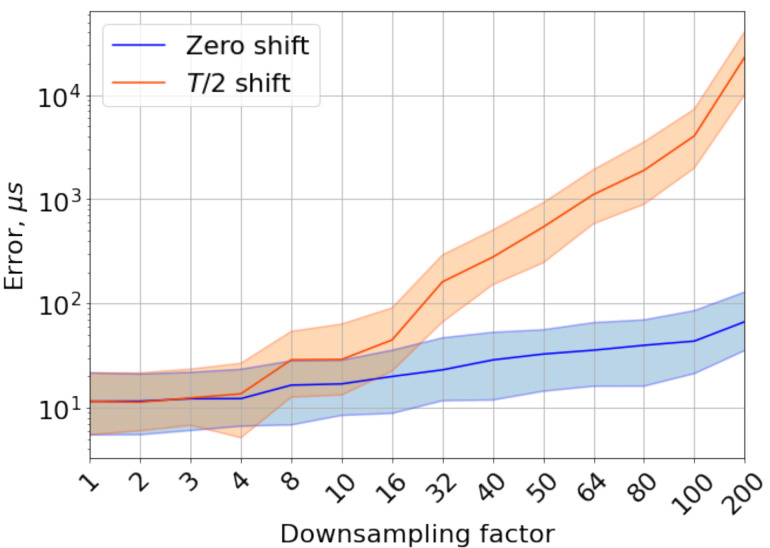
Dependence of error on downsampling factor for 188 Hz cut-off frequency of internal low-pass filter (ILPF), with no additional filtration applied. Two cases of shift between gyroscopes measurements are depicted: zero and half of the sampling period of downsampled data. Bold lines are medians absolute errors; filled areas are interquartile ranges. There is only zero shift for downsampling factor 1 in the graph.

**Figure 7 sensors-21-00068-f007:**
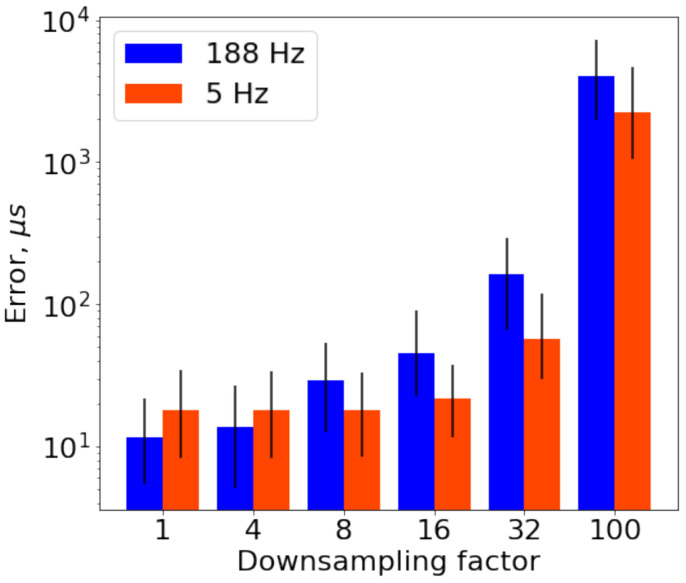
Error dependence on cut-off frequency for two values of cut-off frequencies and several values of downsampling factor. Height of bars are Median Absolute Error (MedAE); length of lines are Interquartile Range (IQR).

**Figure 8 sensors-21-00068-f008:**
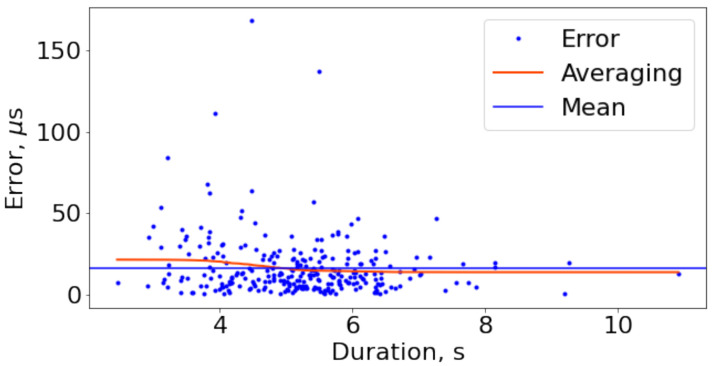
Duration/error dependence statistics of 277 trials with 188 Hz ILPF cut-off frequency. Averaging is carried out by Gaussian filter with deviation of the kernel equals 50. Averaged error (the red line) is slightly higher than the mean value (the blue line) for trials with the shorter duration; at the duration increase, the average lies lower than the mean and has the same value starting from 6 s. Still, there is no strong dependence of error value on the duration of trial.

**Figure 9 sensors-21-00068-f009:**
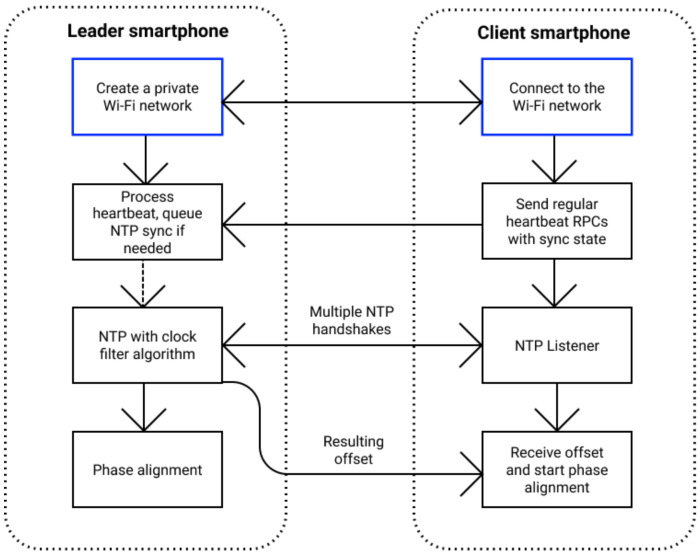
Original libsoftwaresync pipeline.

**Figure 10 sensors-21-00068-f010:**
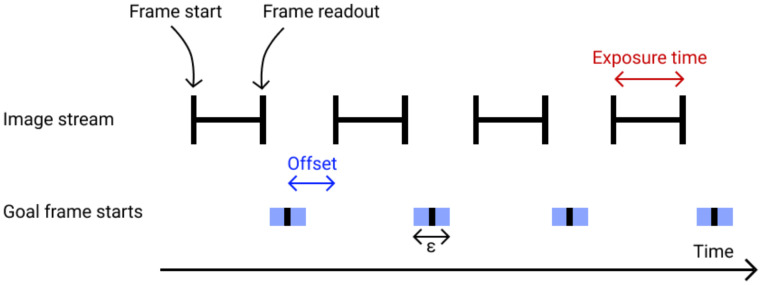
Phase alignment problem setup (the figure is inspired by the scheme from Reference [7]). Application aligns frame capture starts within requested tolerance ϵ.

**Figure 11 sensors-21-00068-f011:**
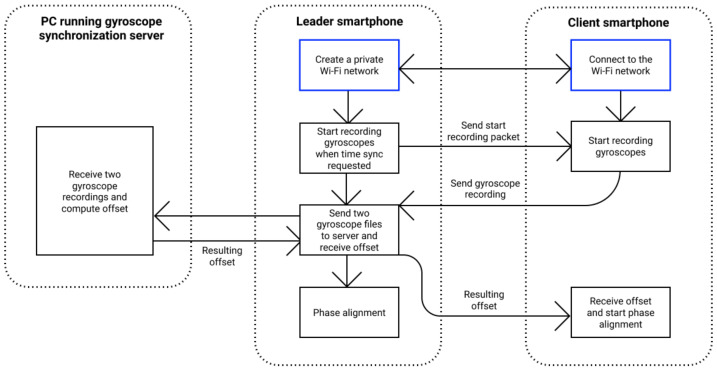
Modified libsoftwaresync pipeline.

**Figure 12 sensors-21-00068-f012:**
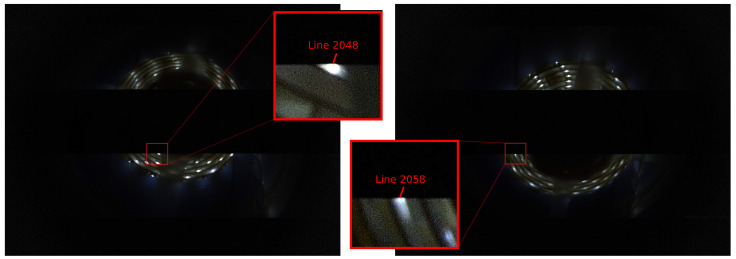
Light source real images obtained during the evaluation. This setup provides the estimation of the offset between two frames shot by different smartphone cameras. The cameras capture a flashing event. Obtained images may have different dark-to-bright edge positions (2048th and 2058th lines in this case). The cameras do not have to be aligned. This simplifies the estimation process compared to that in Reference [7].

**Figure 13 sensors-21-00068-f013:**
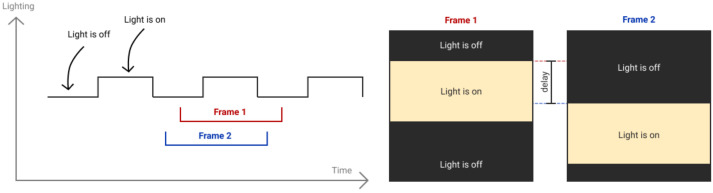
Light source experiment setup.

**Figure 14 sensors-21-00068-f014:**
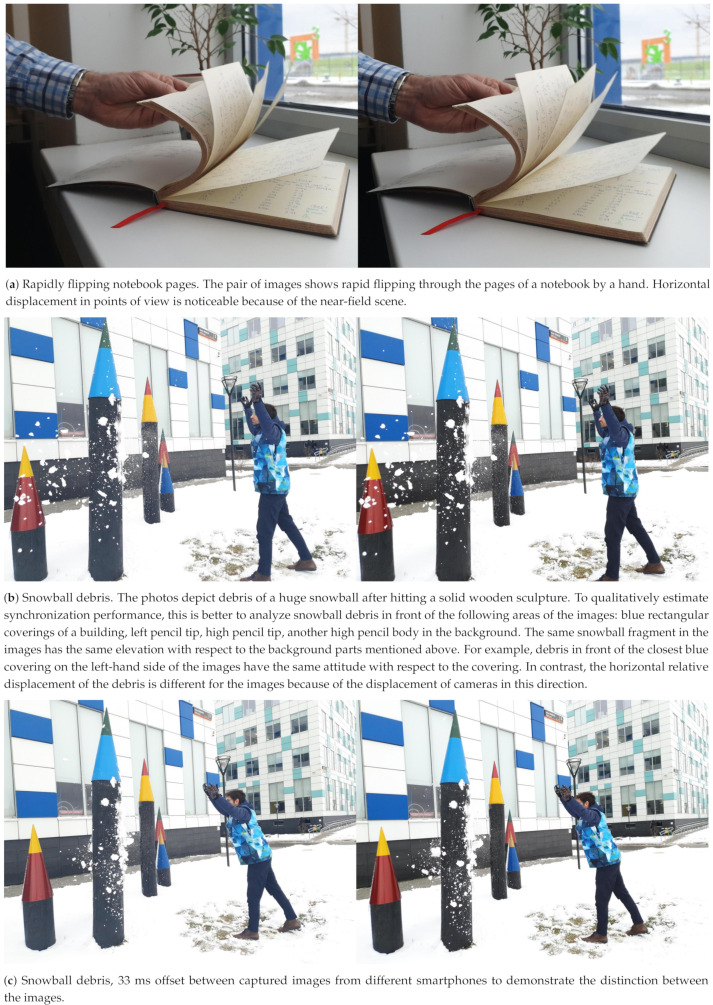
Rapidly flipping notebook pages. The pair of images shows rapid flipping through the pages of a notebook by a hand. Horizontal displacement in points of view is noticeable because of the near-field scene.

**Figure 15 sensors-21-00068-f015:**
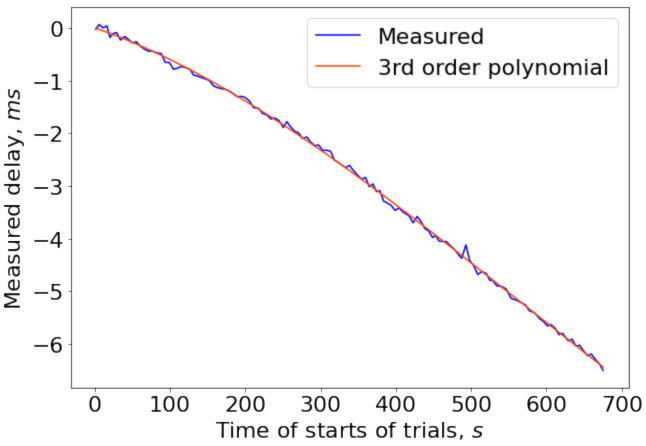
Measured drift between clocks of two smartphones; 3rd order fitted polynomial is also shown to highlight estimation fluctuations.

**Table 1 sensors-21-00068-t001:** Absolute errors in microseconds (μs) for calibrated and not calibrated gyroscopes for 1 kHz sample rate and 188 Hz cut-off frequency.

	MedAE (MAE)	IQR (SD)
W/o calib.	109.67 (164.35)	181.61 (161.97)
With calib.	**11.54 (16.37)**	**16.10 (18.56)**

**Table 2 sensors-21-00068-t002:** A comparison of the performance of Network Time Protocol (NTP)-based and gyroscope-based clock synchronization algorithms performance for synchronous image capture on smartphones.

Method	Mean (μs)	Std (μs)
NTP (by Google Research)	36	22
NTP (by Google Research) with network load	82	27
**Gyro-based with network load**	**16**	**14**

## Data Availability

The data presented in this study are available on request from the corresponding author.

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
