# Peer review of "Twist-n-Sync: Software Clock Synchronization with Microseconds Accuracy Using MEMS-Gyroscopes"

_sensors, 2020, doi:10.3390/s21010068_

Round 1
Reviewer 1 Report
The authors present a software-based strategy to perform time synchronization using gyroscope measurements.
- Please elaborate more on the issues that impede the utilization of hardware-based strategies to perform the time synchronization.
- For technical clarity, the reasons that make quartz oscillators to degrade their performance must be included.
- Does the duration of the procedure used for the trials affect the proposal reliability?
- Describe the procedure used to perform the data frame alignment in the cross-correlation computation.
- Do the gyroscopes used in the proposal have specific requirements? (velocity range among others)
- The paragraph starting in line 297 is confusing. If the authors assume that digital gyroscopes employ sigma-delta ADCs, then the technical foundations of the sigma-delta ADC must be revised. Sigma-delta ADCs use a high sampling frequency rate to mitigate the noise acquired and generated during the conversion.
- How do the authors deal with changing environmental conditions? Gyroscopes have the drift phenomena under changing temperatures.
Reviewer 2 Report
The proposed article is focused issue of synchronization in sensor networks. The authors proposed a new technique for precise software clock synchronization over a network of rigidly attached devices using gyroscope data.
Main comments:
1. Authors should present strict bounds in the resulting estimates, significant for critical systems.
2. What is the applicability of the methods to these systems' unique needs: limited computational cost, the shape of the transmission delays distributions, etc.
3. There is no information about the computational complexity of the whole solution.
Minor comments:
- It would be advisable to include the diagram of the clock synchronization algorithms.
Reviewer 3 Report
Dear authors
I recommend acceptance of this paper due to the innovative ideas and excellent experiment.
question:
- the Chip Scale Atomic Clock (CSAC) is sufficient for this application?
- Does the bias or noise of the gyroscope affect the time syn in this method?
Round 2
Reviewer 1 Report
The authors have addressed the concerns raised by the reviewer